# Histamine and Its Receptors in the Mammalian Inner Ear: A Scoping Review

**DOI:** 10.3390/brainsci13071101

**Published:** 2023-07-20

**Authors:** Lingyi Kong, Ewa Domarecka, Agnieszka J. Szczepek

**Affiliations:** 1Department of Otorhinolaryngology, Head and Neck Surgery, Charité-Universitätsmedizin Berlin, Corporate Member of Freie Universität Berlin, Humboldt-Universität zu Berlin, 10117 Berlin, Germany; lingyi.kong@charite.de (L.K.); ewa.domarecka@charite.de (E.D.); 2Faculty of Medicine and Health Sciences, University of Zielona Gora, 65-046 Zielona Gora, Poland

**Keywords:** histamine, histamine receptors, inner ear, hearing, balance, Ménière’s disease

## Abstract

Background: Histamine is a widely distributed biogenic amine with multiple biological functions mediated by specific receptors that determine the local effects of histamine. This review aims to summarize the published findings on the expression and functional roles of histamine receptors in the inner ear and to identify potential research hotspots and gaps. Methods: A search of the electronic databases PubMed, Web of Science, and OVID EMBASE was performed using the keywords histamine, cochlea*, and inner ear. Of the 181 studies identified, 18 eligible publications were included in the full-text analysis. Results: All four types of histamine receptors were identified in the mammalian inner ear. The functional studies of histamine in the inner ear were mainly in vitro. Clinical evidence suggests that histamine and its receptors may play a role in Ménière’s disease, but the exact mechanism is not fully understood. The effects of histamine on hearing development remain unclear. Conclusions: Existing studies have successfully determined the expression of all four histamine receptors in the mammalian inner ear. However, further functional studies are needed to explore the potential of histamine receptors as targets for the treatment of hearing and balance disorders.

## 1. Introduction

Histamine is a bioactive amine acting as an effector molecule of the immune system and a neurotransmitter in the nervous system. Histamine is synthesized from histidine by specific decarboxylase [1,2]. The histidine decarboxylase-expressing cells include mast cells, basophils, histaminergic neurons, and enterochromaffin-like cells in the stomach [3]. Histamine can either be secreted immediately or stored in granules for later use, as is the case with mast cells and basophils [4,5,6], which release histamine upon stimulation [7,8]. The effects of histamine range from the involvement in innate immunity [9] and its pathologies, such as allergic rhinitis [10], to housekeeping brain and bone homeostasis [4] and depend on the target cell type and the kind of histamine receptor expressed [11].

To date, four histamine receptors have been identified: the H1 receptor (H_1_R), H2 receptor (H_2_R), H3 receptor (H_3_R), and H4 receptor (H_4_R) [12,13]. All four belong to the G-protein-coupled receptor family. H_1_R is expressed in many tissues and cells, including cerebral neurons, the respiratory epithelium, the adrenal medulla, and hepatic, cardiovascular, and endothelial cells [1,14,15]. It is involved in allergic and inflammatory responses. When stimulated, it activates phospholipase C and increases intracellular Ca^2+^ levels [16,17]. As a result, the smooth muscles of the respiratory tract contract, and vascular permeability increases, subsequently causing a range of symptoms associated with allergic reactions [18]. H_2_R is also widely distributed and highly expressed in gastric parietal cells, vascular smooth muscles, the central nervous system, and the heart [15,19,20,21]. H_3_R is mainly found in the central nervous system but it is also widely distributed in peripheral tissues [22]. The effects of H_3_R activation are diverse and include the regulation of histamine turnover, sleep–awake regulation, learning, memory, and inflammation, as well as the inhibition of the release of several other neurotransmitters, such as serotonin, GABA, and glutamate [4,22,23]. The H_4_R was first described at the beginning of the 21st century [24,25,26]. The gene encoding H_4_R was discovered via genomic homology searches and reverse pharmacology, identifying its role in immune and pruritic responses [27]. This receptor has a relatively high homology with H_3_R; however, its role has yet to be elucidated [27,28].

The mammalian inner ear is a complex sensory organ consisting of the cochlea, vestibule, and three semicircular canals [29]. Sound passes from the external ear canal through the middle ear to the inner ear. In the inner ear’s cochlea, sound’s mechanical energy is converted into a biochemical signal by the sensory epithelium (hair cells) in a process called mechanotransduction. Mechanotransduction induces the release of glutamate from the inner hair cells, which activates the spiral ganglion neurons by initiating an action potential that is sequentially transmitted along the auditory pathway to the auditory cortex [30,31]. 

The vestibular system is responsible for sensing and processing information about the position and movement of the head and body in space and maintaining balance and coordination during the movement [32]. The peripheral vestibular organs are located bilaterally in the inner ear. They consist of two otolithic organs (utricle and saccule) and three semicircular canals (anterior, posterior, and horizontal), the former sensing linear acceleration, such as head movement or gravity, and the latter sensing rotational acceleration [33]. The sensory nerve epithelium in the utricle and saccule is the macula, and in the semicircular canals, it is the crista ampullaris [32]. Both structures contain vestibular hair cells, which release glutamate upon depolarization, stimulating the vestibular ganglion’s afferent nerves. The vestibular ganglion and the cochlear spiral ganglion neurons form the eighth cranial nerve.

The endolymphatic sac is the non-sensory part of the membranous labyrinth of the inner ear. It plays several important roles, including regulating the volume and pressure of the potassium-rich endolymph fluid, participating in the immune response within the inner ear, and removing waste products from the endolymph. 

In recent years, the immune function, inflammatory processes, and vascular control of the inner ear have been investigated and reviewed [34,35,36,37]. However, the specific topic of histamine and its signaling in the cochlea or vestibular organs remains scarcely addressed in the literature. A few research teams have identified the expression of histamine receptors in the inner ear of mammals [38,39,40,41]. Additionally, in 2020, our group found mast cells in the inner ear of both rats and mice [42]. Mast cells are the major source of histamine in the body, along with basophils, gastric parietal cells, and the central nervous system [43]. Upon activation, mast cells degranulate and release a number of immuno- and neuromodulatory compounds, including histamine [44,45]. Some conditions necessary for mast cell activation have already been described regarding the inner ear, including IgE antibody transcytosis across the blood–labyrinth barrier [46] and the presence of substance P [47]. However, more research is needed to understand the relationship between the presence of mast cells in the inner ear, their mode of activation, potential histamine release, and its consequences in health and disease, such as mastocytosis or IgE-mediated diseases. Clinical evidence suggests an association between elevated numbers of mast cells and inner ear disorders [48,49]. Moreover, experiments demonstrated that Meniere’s disease-like symptoms (attacks of nystagmus and hearing loss) can be induced by the experimental induction of a type I allergy in the endolymphatic sac of guinea pigs [50]. 

Therefore, the purpose of this review is to provide an overview of current research investigating the expression and function of histamine and its receptors in the mammalian inner ear. A further aim is to assess the potential of the identified research for developmental, neuroprotective, and clinical applications in the inner ear. By reviewing the literature, we aim to systematically map the research that has been conducted in this area and chart the course for future research.

## 2. Materials and Methods

### 2.1. Search Strategy and Databases

This scoping review followed the Preferred Reporting Items for Systematic Review and Meta-Analyses (PRISMA) statement [51]. A systematic search was conducted by a reviewer using the following databases: PubMed, Web of Science, and OVID EMBASE. The search window covered the period between January 2000 and June 2023. Redefined search strategies and selection criteria were used to evaluate the eligibility of studies. The keywords included the following combination of MeSH terms: histamine AND ((inner ear) OR (cochlea*)). We identified 181 publications through the database search and reference lists. The references were exported to Rayyan, a citation management system (https://rayyan.ai/, accessed on 1 June 2023). After removing duplicates, 87 articles remained in the database. 

### 2.2. Screening Process and Inclusion and Exclusion Criteria 

Two independent reviewers screened titles and abstracts of the remaining 87 articles. Criteria for the inclusion of studies in this review were English publications and studies on histamine (receptors) relevant to the mammalian inner ear or cochlea. Exclusion criteria were review articles, case reports on other topics, the lack of an English version, and non-mammalian studies. Articles not meeting the inclusion criteria and deemed irrelevant to the study were excluded. 

### 2.3. Study Eligibility and Data Extraction

Twenty-eight publications were included for full-text eligibility assessment. After excluding publications for which the full text could not be found and research publications with non-primary research content, 18 publications were finally included in the discussion of this review. Data were extracted from each included publication to collate study characteristics and details. The PRISMA flowchart for this study is presented in Figure 1.

## 3. Results

### 3.1. Characteristics of Included Studies

A thorough evaluation of studies published between 2000 and 2023 was performed. We classified eligible publications as histamine receptor expression or functional studies in the mammalian inner ear. Nine publications investigated the expression of histamine receptors in the inner ear. The sites of expression, receptor subtypes, and species studied in the articles varied and are summarized in Table 1. 

Eleven publications dealt with the functional investigation of histamine receptors. The purpose of the studies varied, including pathophysiology, pharmacological treatment, or predictive purposes, as shown in Table 2. Some studies used histamine receptor agonists or antagonists as useful tools for indirect studies of histamine function. Of the eighteen articles included in our review, eight (44.4%) used agonists or antagonists, and six of these eight studies (75%) used the H_3_R antagonist betahistine. We further summarized the evidence from the studies and discussed the functional context of the findings. 

The main areas covered by the included studies were the anatomical/cellular localization of histamine receptors, vascular permeability, and electrophysiology. However, only one of the studies covered more than one area, combining the anatomical and electrophysiological approaches (Figure 2). One publication, which was not included in the Venn diagram [52], dealt with cochlear metabolomics.

**Figure 2 brainsci-13-01101-f002:**
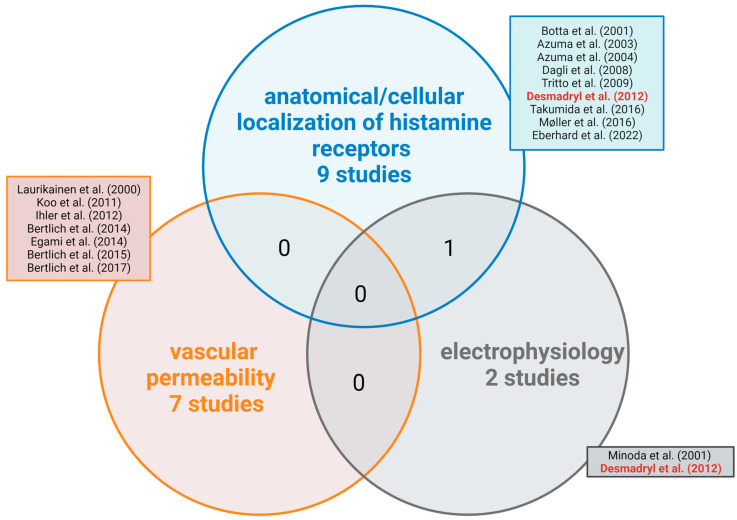
Venn diagram of the main areas of coverage of the studies included in this review [38,39,40,41,53,54,55,56,57,58,59,60,61,62,63,64,65]. Red font indicates the only publication covering two areas. All publications are listed and referenced in Table 2. The study of Wang et al. was omitted in this figure as it was the only study representing metabolomics [52]. Created with BioRender.com.

**Table 2 brainsci-13-01101-t002:** Characteristics of included studies.

Author and Year	Location of Study	Species	Age or Weight	Purpose of the Study	Involved Histamine Receptors				Involved Agonists or Antagonists	Inner Ear Structures Focused on	Main Finding(s)
Laurikainen et al. (2000) [58]	Finland	Guinea pig	-	Function-related	1		3		Yes (betahistine)	Cochlea	The effect of intravenous betahistine on cochlear blood flow was found to be dose-dependent and greater in the cochlear vasculature than in the systemic vascular bed, as measured with laser Doppler.
Minoda et al. (2001) [59]	Japan	Guinea pig (Dunkin–Hartley)	300–400 g	Function-related	1	2			Yes (pyrilamine and Cimetidine)	Cochlea	In low concentrations, histamine may act as an extracellular signal on inner hair cells or it may stimulate the afferent nerve by binding to their H_1_R and H_2_R.
Azuma et al. (2003) [38]	Japan	Rat (Wistar)	200–280 g	Expression-related	1	2	3		No	Cochlea (modiolus)	H_1_, H_2_, and H_3_R mRNA were detected in the rat cochlear modiolus.
Azuma et al. (2004) [39]	Japan	Rat (Wistar)	200–280 g	Expression-related	1	2	3		No	Cochlea (spiral ganglion)	H_1_R, H_2_R, and H_3_R staining was observed in spiral ganglion cells in rat cochlea. Neurofilament-200 staining indicated that HR expression is specific to type I ganglion cells.
Botta et al. (2008) [40]	Italy	Mouse (C57BL/6)	3 weeks	Expression-related	1		3		No	Vestibule (semicircular canal)	Mouse vestibular epithelia express H_1_R. Conversely, no clear evidence for H_3_R expression was found.
Dagli et al. (2008) [41]	Turkey	Rabbit (New Zealand)	2–3 kg	Expression-related	1	2	3		No	Endolymphatic sac	The endolymphatic sac of rabbits was positive for H_1_R, H_2_R and H_3_R. Cells positive for all receptors were found in the epithelial and subepithelial layers of the duct and the proximal endolymphatic sac.
Tritto et al. (2009) [53]	Italy	Mouse (Swiss CD-1)	3 weeks	Expression-related			3		No	Vestibule (vestibular ganglion)	Calyx and dimorphic neurons of mouse Scarpa’s ganglion express H_3_R.
Koo et al. (2011) [60]	USA	Mouse (C57BL/6)	4–7 weeks	Function-related					No	Cochlea (the organ of Corti)	Systemic increases in serum levels of vasoactive peptides, like histamine, can modulate cochlear uptake of gentamicin, likely via permeability changes in the blood–labyrinth barriers.
Desmadryl et al. (2012) [54]	France	Rat (Wistar or Long–Evans)	2–8 days	Expression- and function-related			3	4	Yes (JNJ 10191584; JNJ 7777120; 4-methylhistamine; thioperamide; betahistine)	Vestibule (vestibular ganglion)	H_4_ and H_3_R transcripts are present in the rat vestibular ganglion, and H_4_R antagonists have a significant inhibitory effect on vestibular neuron activity.
Ihler et al. (2012) [61]	Germany	Guinea pig (Dunkin–Hartley)	250–400 g	Function-related	1		3		Yes (betahistine)	Cochlea (stria vascularis)	The improved effects of higher doses of betahistine in the treatment of Meniere’s disease might be due to a corresponding increase in cochlear blood flow.
Bertlich et al. (2014) [62]	Germany	Guinea pig (Dunkin–Hartley)	2–8 weeks	Function-related	1		3		Yes (betahistine)	Cochlea	Aminoethylpyridine and hydroxyethylpyridine are, like betahistine, able to increase cochlear blood flow significantly. The effect of aminoethylpyridine was greatest. Pyridylacetic acid had no effect on cochlear microcirculation.
Egami et al. (2014) [63]	Japan	Guinea pig (Dunkin–Hartley)	250–300 g	Function-related	1				Yes (olopatadine hydrochloride)	Vestibule (endolymphatic sac)	The systemic sensitization with DNP-As produced allergy-induced experimental endolymphatic hydrops with type 1 hypersensitivity allergic reaction, and the development of these endolymphatic hydrops was prevented by H_1_R antagonists.
Bertlich et al. (2015) [64]	Germany	Guinea pig (Dunkin–Hartley)	180–300 g	Function-related	1		3		Yes (betahistine)	Cochlea	Betahistine affects cochlear blood flow through histaminergic H_3_ heteroreceptors.
Takumida et al. (2016) [56]	Japan	Mouse (CBA/J)	10 weeks	Expression-related	1	2	3	4	No	Cochlea and vestibule (the organ of Corti, spiral ganglion, vestibular ganglion, endolymphatic sac, macula)	All four types of histamine receptors are present in the inner ear.
Møller et al. (2016) [55]	Denmark	Human	-	Expression-related	1	2	3	4	No	Vestibule (endolymphatic sac)	The H_1_R was expressed in the epithelial lining of the endolymphatic sac, whereas H_3_R was expressed exclusively in the subepithelial capillary network. H_2_R and H_4_R were not expressed.
Bertlich et al. (2017) [65]	Germany	Guinea pig (Dunkin–Hartley)	200–450 g	Function-related	1		3		Yes (betahistine)	Cochlea (stria vascularis)	Betahistine has an active effect on cochlear microcirculation, and its main mode of action is evidently active dilation of precapillary arterioles.
Eberhard et al. (2022) [57]	Denmark	Human	48–69 years	Expression-related	1		3		No	Vestibule (saccule)	Intense expression of the H_3_R was found in the non-sensory epithelial lining cells of the human saccule, whereas there was no H_1_R expression.
Wang et al. (2022) [52]	China	Mouse (C57BL/6)	4 and 48 weeks	Function-related					No	Cochlea	This study represents the first comprehensive comparison of cochlear metabolites between young and old mice, revealing significant upregulation of histamine as a notable metabolic change in the cochlea of the old C57BL/6 group.

### 3.2. Expression of Histamine Receptors in the Mammalian Cochlea

Results from several studies have shown that histamine receptors were detected at multiple sites in the mammalian inner ear (Table 1, Figure 3) [38,39,56]. The mRNA encoding histamine receptors (H_1_, H_2_, and H_3_R) was found to be present in the lateral portion of the cochlea, including the spiral ligament and the stria vascularis, the medial portion, including the organ of Corti, and the modiolus [38]. Takumida et al. used immunohistochemistry (IHC) to examine the mouse cochlea and found that the stria vascularis was positive for H_1_R, the spiral ligament for H_3_R and H_4_R, and the spiral ganglion neurons for all four types of receptors. In the latter, immunofluorescence (IF) receptor signals were observed in the cytoplasm of the spiral ganglion neurons [56]. In the organ of Corti, H_3_R was observed in both the outer and inner hair cells, whereas H_1_R, H_2_R, H_3_R, and H_4_R were observed in some supporting cells [56].

### 3.3. Expression of Histamine Receptors in the Mammalian Vestibular System

#### 3.3.1. Semicircular Canals

Using reverse RT-PCR, Western blotting, and in situ immunolabeling, Botta et al. demonstrated that the semicircular canal of the mouse expresses H_1_R [40]. Conversely, no clear evidence for H_3_R expression was found.

#### 3.3.2. Utricle and Saccule

All four types of histamine receptors have been demonstrated in the maculae of the saccule and utricle [53,56]. More specifically, the expression of histamine receptors was found in type I hair cells, the calyx and dimorphic vestibular afferents, and subepithelial cells [53]. 

#### 3.3.3. Vestibular Ganglion

Like in spiral ganglion neurons, H_1_R, H_2_R, H_3_R, and H_4_R were detected on vestibular ganglion cells using immunohistochemistry [53,54,56]. Tritto et al. examined vestibular neurons with immunofluorescence and found that approximately 30% of the nerves stained positively for H_3_R [53]. 

#### 3.3.4. Endolymphatic Sac

In the murine endolymphatic sac, H_1_R, H_2_R, H_3_R, and H_4_R were detected on the epithelial cells [56]. H_1_R, H_2_R, and H_3_R were also found in the epithelial and subepithelial layers of the ducts and proximal endolymphatic sac of rabbits [41]. It has been shown that H_3_R is highly expressed in non-sensory epithelium [57], suggesting that it may play a role in maintaining cochlear fluid homeostasis. Histamine receptor expression was also found in the human endolymphatic sac. cDNA microarray data and immunohistochemical staining revealed there the presence of H_1_R and H_3_R proteins and transcripts [55]. Additionally, H_1_R was found in the endolymphatic sac lining, whereas H_3_R was present in the subepithelial capillary network [55]. 

## 4. Discussion

This scoping review evaluated the current literature concerning the expression and function of histamine and its receptors in the mammalian inner ear. All four types of histamine receptors have been found to be expressed in multiple sites of the inner ear, where they act in coordination to maintain auditory processing and balance. Therefore, it is tempting to speculate that histamine receptors might have an impact on the maintenance of inner ear function. The functional role of histamine and its receptors in the inner ear in the context of the included publications is discussed below.

### 4.1. Histamine Alters Vascular Permeability

One of the major physiological functions of histamine found in the inner ear is the ability to alter vascular permeability, primarily depending on H_1_R located on endothelial cells of the inner ear vascular lining [60,66]. A study found that histamine disrupts endothelial barrier formation in microvenules, as evidenced by changes in the localization of vascular endothelial cadherin (VE-cadherin) at endothelial cell junctions, and these manifestations can be eliminated by using H_1_R antagonists [66]. An increase in vascular permeability can produce beneficial physiological effects. For example, when tissue is injured or infected, an increase in vascular permeability allows white blood cells and antibodies to move out of the bloodstream and into the affected area, where they can help fight off the infection and promote healing [67]. On the other hand, an increase in vascular permeability can also help deliver nutrients and oxygen to damaged tissues, which is essential for repair and regeneration [68]. However, an excessive release of histamine can also lead to pathological effects. Koo et al. showed that vasoactive peptides, such as histamine, can enhance the ototoxicity of aminoglycosides by altering the permeability of the blood–labyrinth barrier in the mouse cochlea [60]. Like the blood–brain barrier, the blood–labyrinth barrier is composed of specialized cells and tight junctions that prevent the entry of large molecules, immune cells, and other potentially harmful substances into the inner ear from the bloodstream. These observations suggest an increased risk of ototoxicity during bacterial infections during aminoglycoside therapy, with adverse consequences for hearing function during recovery. A further exploration of this issue can reveal how to circumvent side effects and mitigate unnecessary hearing damage.

### 4.2. Electrophysiological Studies—The Role of Histamine in the Transmission of Electrical Signals of Sound

Only two studies have studied the electrophysiological function of histamine in the mammalian inner ear [54,59]. Previous results obtained using the vestibular organ of frogs and lateral line of Xenopus laevis have suggested that histamine may act as a hair cell transmitter [69,70,71]. These studies provided evidence that histamine increases the afferent firing rate in nerves and that this afferent firing could be blocked by H_1_R and H_2_R antagonists. Regarding guinea pigs, Minoda et al. reported that the infusion of histamine at low concentrations (10 and 50 μM) increased the compound action potential (CAP) amplitude without affecting the cochlear microphonic (CM), and the increase in CAP amplitude could be suppressed by H_1_R and H_2_R antagonists (50 μM) [59]. CAP represents the synchronous discharge of many cochlear afferent nerve fibers and is an indirect indicator of afferent nerve fiber activity. This result confirms previous findings in non-mammals. Therefore, histamine may act as an extracellular stimulatory signal that influences sound signaling via H_1_R and H_2_R in the cochlea.

Chávez et al. examined the effect of H_3_R agonists and antagonists on afferent neuron electrical discharge in the isolated inner ear of the axolotl [72]. They found that H_3_R antagonists inhibit the nerve afferent discharge in the semicircular canals. It can be hypothesized that the antivertigo action of histamine-related drugs may be caused, at least in part, by a decrease in the sensory input from the vestibular end organs. However, there are varying opinions on this matter. Earlier work regarding the semicircular canals of a frog showed that betahistine significantly reduced the resting, but not the afferent, discharge [71,73]. Unfortunately, this has not been confirmed in the mammalian vestibule. There are relatively few electrophysiological studies on H_4_R. Desmadry et al. found that the antagonism of H_4_R leads to a strong reversible inhibition of evoked action potential firing [54]. Moreover, the systemic in vivo administration of 4-methylhistamine, an H_4_R antagonist, effectively reduces the vestibular behavioral deficits induced by peripheral injury [54]. This demonstrates the potential role of the H_4_R as a pharmacological target for the treatment of vestibular disease.

### 4.3. Histamine May Affect Hair Cell Synaptic Transmission by Binding to Histamine 3 Receptor

H_3_R was originally described as an autoreceptor, inhibiting the release of histamine from histaminergic neurons in the brain [4]. H_3_R was shown to modulate inflammatory processes in the brain and the properties of neuronal synapses and has also been associated with the emergence of neurodevelopmental disorders [4]. Recent evidence suggests that the H_3_R is a pre- and postsynaptic receptor, regulating the release of several important neurotransmitters (such as acetylcholine, dopamine, GABA, norepinephrine, and serotonin) both in the peripheral and central nervous systems [4,22]. In the mammalian inner ear, the H_3_R has already been detected in many locations along the vestibulocochlear pathway, including the spiral ganglion and vestibular ganglion, the stria vascularis, and the endolymphatic sac [38,39,41,56]. However, it is worth mentioning that in the organ of Corti of the adult mouse, hair cells and supporting cells were also found to express H_3_R [56]. Glutamate is the main neurotransmitter at the hair cell afferent synapse [74]. Studies have found that histamine causes glutamate release from cultured astrocytes [4]. In addition, other studies have demonstrated that Ciproxifan, an H_3_R antagonist, presynaptically inhibits glutamate release in the rat hippocampus [75]. Although previous studies have confirmed the presence of H_3_R in the inner ear, its specific role in the regulation of afferent signaling pathways remains to be elucidated.

### 4.4. Clinical Application

The endolymphatic sac is a non-sensory segment of the inner ear and a part of the membranous labyrinth [76]. Its main function is to maintain the fragile endostasis of the endolymphatic and ectolymphatic vessels and to remove endolymphatic waste products [76,77]. One of the most common pathologies of the endolymphatic compartment (cochlear duct, also known as scala media) is endolymphatic hydrops. In this condition, characteristic for Menière’s disease, the excessive pressure in the cochlear duct ruptures physical barriers of the endolymphatic space causing a temporary loss of hearing and vestibular function [65,78]. The proposed causes of endolymphatic hydrops appear to be heterogeneous. Interestingly, experiments in the guinea pig have shown that histamine, released from the mast cells of the endolymphatic sac, induces a calcium response in the vestibular hair cells that is mediated by H_1_R, H_2_R, and H_3_R [41,79]. The histamine-mediated vasodilation of the endolymphatic sac could also lead to the deposition of immune complexes and endolymphatic effusion. These histamine-induced functional changes may be involved in the pathophysiology of Ménière’s disease.

Betahistine, a structural analog of histamine, is a weak H_1_R agonist and a strong H_3_R antagonist [64,80]. It is used to treat Ménière’s disease, particularly in central Europe [78]. Clinical trials found that repetitive daily doses of betahistine reduce the number and severity of attacks during the course of the disease [61,64]. Bertlich et al. studied the effects of betahistine on cochlear pericapillary cells and precapillary arteries and showed that the main mode of action was apparently the active dilation of the precapillary arteries [64]. Some researchers have suggested that the dilation may be due to the activation of H_1_R and H_2_R on the cochlear vasculature. The H_1_R and H_2_R expressed in the vascular smooth muscles contribute to vascular contraction and dilation [81]. A subsequent study examined changes in cochlear blood flow and blood pressure by separately blocking specific histamine receptors and found that the activation of H_3_R caused the decrease in cochlear blood flow and blood pressure, rather than H_1_R or H_2_R [64]. The infusion of the histaminergic H_3_R antagonist thioperamide prior to betahistine infusion completely reversed the effects of betahistine on cochlear blood flow [58]. 

Two main mechanisms of action of betahistine have been proposed. One is the counter-antagonistic effect of betahistine on H_3_R, which is thought to contribute to central neural compensation in the presence of peripheral vestibular imbalance [64]. The second involves an enhanced cochlear microcirculation in the stria vascularis [61,62,64]. However, betahistine may also cause clinical adverse effects, including flushing, headache, skin reactions, and hypotension, which are typical of H_1_R-related reactions and challenge the selectivity of the drug. The current findings provide a vision for future studies to optimize drug efficacy, for example, by targeting a specific symptom of Ménière’s disease by reducing drug side effects while maintaining efficacy.

### 4.5. Limitations

In the brain, researchers have found that histamine could be a disruptor of neurodevelopment [4,82]. Histamine is detected early in brain development and may therefore contribute to brain formation by regulating processes such as neuronal outgrowth, synaptogenesis, neuronal differentiation, and migration [83,84]. However, high levels of histamine can adversely affect neurodevelopment [83,84]. Suppressing neurogenesis in early development can lead to cognitive deficits and various developmental disorders. 

The experimental specimens in the studies reviewed in this review were primarily from adult mammals. The expression of histamine receptors in different regions of the adult mammalian inner ear was confirmed. A quasi-targeted metabolomic analysis also indicates that the elevation of histamine in the inner ear is one of the targeted metabolic changes in age-related hearing loss [52]. However, the expression pattern of histamine receptors in the developing cochlea and the physiological role of these receptors remain unknown, and we did not identify any studies on this topic. Significant histamine receptor expression in the spiral and vestibular ganglia has been demonstrated in the available studies. However, it is still unclear when these receptors are first expressed and whether they impact the maturation of hearing and the maintenance of homeostatic function. This knowledge gap opens a field for further research.

### 4.6. Future Directions

The first imposing direction for future research on histamine and the inner ear is one that takes into account experiments on cochlear mast cell degranulation and its effects on inner ear morphology and physiology. In fact, some of these experiments are already underway in our laboratory and promise to provide some answers to the question of the effect of mast cell degranulation on cochlear tissue.

Furthermore, while processing the data for this review, we noticed a lack of research on the role of histamine and its receptors in inner ear development. Therefore, one of the goals of future research could be to investigate the dependence of inner ear development on the histaminergic system. 

Another avenue of research could be to study the function of histamine in inner ear hair cells. Of particular interest would be the H_3_R, which has been found to be expressed by both outer and inner hair cells in the mouse cochlea [56], suggesting that histamine could potentially affect their function. 

Finally, a thorough confirmation of histamine receptor expression in the human inner ear is needed to justify further translational research. This may be possible through initiatives such as the National Temporal Bone Registry established by the Massachusetts Eye and Ear, a teaching hospital of Harvard Medical School in Boston. In addition, pharmacologic safety monitoring of histamine receptor agonists and antagonists requires special attention to the peripheral auditory and vestibular systems.

## 5. Conclusions

This scoping review provides an overview of the current knowledge of histamine and its receptors in the mammalian inner ear. It highlights the evidence and potential functional significance of the presence of histamine and its receptors in hearing and balance. We have also noted that histamine-receptor-related agonists and antagonists are readily used to study histamine receptors and contribute to the treatment of Ménière’s disease. Many histamine-related drugs have been used clinically to treat otologic disorders, but research into specific mechanisms has not yet provided clear answers. This finding underscores the importance of further research into the complex molecular mechanisms of histamine and its receptors involved in auditory and vestibular function. Further research is also needed to fully understand the role of histamine receptors in inner ear development.

## Figures and Tables

**Figure 1 brainsci-13-01101-f001:**
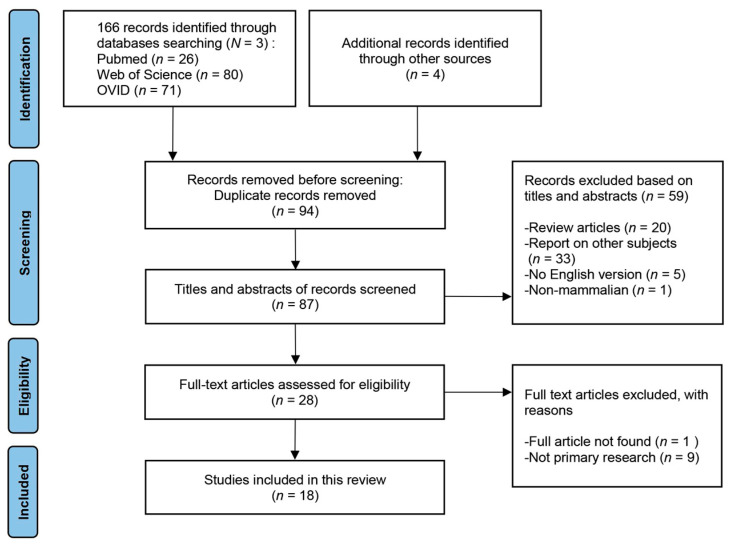
PRISMA study flow diagram. “*N*” signifies the number of databases; “*n*” signifies the number of publications.

**Figure 3 brainsci-13-01101-f003:**
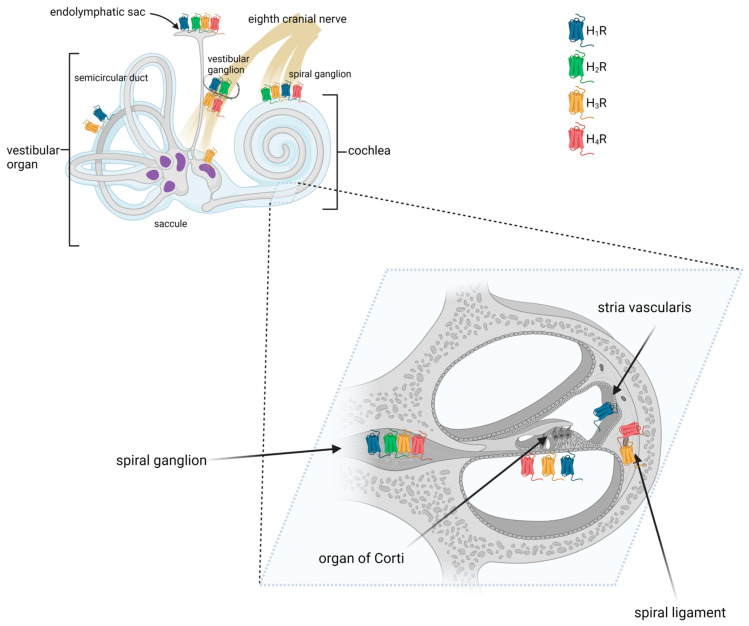
Distribution of histamine receptors in the mammalian inner ear. Created with BioRender.com.

**Table 1 brainsci-13-01101-t001:** Sites and subtypes of histamine receptor expression in the mammalian inner ear.

Author and Year	Species	Age or Weight	Experimental Methods	H_1_R	Locations	H_2_R	Locations	H_3_R	Locations	H_4_R	Locations
Azuma et al. (2003) [38]	Rat (Wistar)	200–280 g	PCR	✓	modiolus	✓	modiolus	✓	modiolus	n.d.	
Azuma et al. (2004) [39]	Rat (Wistar)	200–280 g	IHC	✓	spiral ganglion	✓	spiral ganglion	✓	spiral ganglion	n.d.	
Botta et al. (2008) [40]	Mouse (C57BL/6)	3 weeks	PCR; IF	✓	semicircular canal	n.d.		✓	semicircular canal	n.d.	
Dagli et al. (2008) [41]	Rabbit (New Zealand)	2000–3000 g	IHC	✓	endolymphatic sac	✓	endolymphatic sac	✓	endolymphatic sac	n.d.	
Tritto et al. (2009) [52]	Mouse (Swiss CD-1)	3 weeks	PCR; IF	n.d.		n.d.		✓	vestibular ganglion	n.d.	
Desmadryl et al. (2012) [53]	Rat (Wistar)	2–8 days	PCR	n.d.		n.d.		✓	vestibular ganglion	✓	vestibular ganglion
Møller et al. (2016) [54]	Human	n.s.	IHC, microarrays	✓	endolymphatic sac (IHC, microarrays)	✓	endolymphatic sac (microarrays, expression greater than in dura)	✓	endolymphatic sac (IHC, microarrays)	✓	endolymphatic sac (microarrays, expression lesser than in dura)
Takumida et al. (2016) [55]	Mouse (CBA/J)	10 weeks (25–30 g)	IHC; PCR	✓	stria vascularis; the organ of Corti; spiral ganglion; vestibular ganglion; vestibular epithelium; endolymphatic sac	✓	spiral ganglion; vestibular ganglion; vestibular epithelium; endolymphatic sac	✓	spiral ligament; the organ of Corti; spiral ganglion; vestibular ganglion; vestibular epithelium; endolymphatic sac	✓	spiral ligament; the organ of Corti; spiral ganglion; vestibular ganglion; vestibular epithelium; endolymphatic sac
Eberhard et al. (2022) [56]	Human	48–69 years	IHC	neg.	saccule	n.d.		✓	saccule	n.d.	

Abbreviations: PCR, polymerase chain reaction; IF, immunofluorescence; IHC, immunohistochemistry. “✓”—indicates the confirmed presence of a given histamine receptor; “n.d.”—not done; “n.s.” not stated; “neg.”—negative results.

## Data Availability

Not applicable.

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
