# Peer review of "Histamine and Its Receptors in the Mammalian Inner Ear: A Scoping Review"

_brainsci, 2023, doi:10.3390/brainsci13071101_

Round 1

Reviewer 1 Report

Good job. I suggest including future directions.

Author Response

We thank the Reviewer for taking the time to read and review our manuscript. Reviewer’s constructive criticism is greatly appreciated, and implementing the pitched ideas improved the readability and quality of our work.

Below you will find the reply to the comment made.

I suggest including future directions.

We followed this suggestion and added the following paragraph to the Discussion:

4.6 Future directions

The first imposing direction for future research on histamine and the inner ear is one that takes into account experiments on cochlear mast cell degranulation and its effects on inner ear morphology and physiology. In fact, some of these experiments are already underway in our laboratory and promise to provide some answers to the question of the effect of mast cell degranulation on cochlear tissue.

Furthermore, while processing the data for this review, we noticed a lack of research on the role of histamine and its receptors in inner ear development. Therefore, one of the goals of future research could be to investigate the dependence of inner ear development on the histaminergic system.

Another avenue of research could be to study the function of histamine in inner ear hair cells. Of particular interest would be the H3R, which has been found to be expressed by both outer and inner hair cells in the mouse cochlea[53], suggesting that histamine could potentially affect their function.

Finally, confirmation of histamine receptor expression in the human inner ear is needed to justify further translational research. This may be possible through initiatives such as the National Temporal Bone Registry established by the Massachusetts Eye and Ear, a teaching hospital of Harvard Medical School in Boston. In addition, pharmacologic safety monitoring of histamine receptor agonists and antagonists requires special attention to the peripheral auditory and vestibular systems.

Reviewer 2 Report

A large part of hearing research involves the topic of hearing loss as a result of aging, high-intensity noise exposure or ototoxic damage. Often there is evidence of hair cell damage directly related to a mechanical insult but much of the damage is more subtle and is thought to involve neuroinflammatory processes. In most peripheral tissues histamine has an important role in inflammation but comparatively little work has been done on histamine release in the inner ear. This review provides a timely synthesis of the available literature and identifies particular areas where there is a need for further research to improve our understanding of the role of histamine in pathological processes in the inner ear. The review is logically arranged and clearly written in a concise, direct way. There is an extensive literature on the use of betahistine in the treatment of Meniere’s disease, but I think the authors are wise not to include these studies in their final sample. They do discuss some of the recent work summarizing the betahistine work and I think this is adequate. There are a few unnecessary phrases or typographical mistakes which I indicate below. Otherwise, I can’t think of any obvious improvements for the manuscript.

 Minor corrections:

Page 2 line 55 I would replace “the endolymphatic sac”  with “the semicircular canals”. The canals are an important part of the sensory apparatus while the sac is not.

Page 2 lines 72 and 73  I would miss this sentence out. It doesn’t add anything and is misleading. The inner ear is not part of the periphery of the brain. The inner ear forms part of the peripheral nervous system while the brain forms the central nervous system.

Page 3 line 107 I don’t understand the significance of the superscript 1. It looks like a reference to a footnote but I can’t see a footnote.

Table 2 Laurikainen reference last column should be about betahistine not “batahistine”

Generally very good. There a few instances of the definite article being omitted but these are minor and I didn't note them down.

Author Response

We thank the Reviewer for taking the time to read and review our manuscript. Reviewer’s constructive criticism is greatly appreciated, and implementing the pitched ideas improved the readability and quality of our work.

Below you will find point-to-point replies to all comments made.

Page 2 line 55 I would replace “the endolymphatic sac”  with “the semicircular canals”. The canals are an important part of the sensory apparatus while the sac is not.

We revised the sentence in line 55 in the suggested way:

The mammalian inner ear is a complex sensory organ consisting of the cochlea, vestibule, and three semi-circular canals [29].

Page 2 lines 72 and 73  I would miss this sentence out. It doesn’t add anything and is misleading. The inner ear is not part of the periphery of the brain. The inner ear forms part of the peripheral nervous system while the brain forms the central nervous system.

The sentence in question was removed, and the entire paragraph was entirely rewritten:

The vestibular system is responsible for sensing and processing information about the position and movement of the head and body in space and maintaining balance and coordination during the movement [32]. The peripheral vestibular organs are located bilaterally in the inner ear. They consist of two otolithic organs (utricle and saccule) and three semicircular canals (anterior, posterior, and horizontal), the former sensing linear acceleration, such as head movement or gravity, and the latter sensing rotational acceleration [33]. The sensory nerve epithelium in the utricle and saccule is the macula, and in the semicircular canals is the crista ampullaris [32]. Both structures contain vestibular hair cells, which release glutamate upon depolarization, stimulating the vestibular ganglion's afferent nerves. The vestibular ganglion and the cochlear spiral ganglion neurons form the eighth cranial nerve.

Page 3 line 107 I don’t understand the significance of the superscript 1. It looks like a reference to a footnote but I can’t see a footnote.

Thank you for pointing this out – it used to be a footnote; we removed it.

Table 2 Laurikainen reference last column should be about betahistine not “batahistine”

The mistake was corrected.

Reviewer 3 Report

The authors searched three databases (Pubmed, WoS and Embase) for histamine+inner ear articles from year 2000 and onwards and found 18 useful publications on the four histamine receptor types, their locations and possible chemical effects. Although many things remain to be explained in this field the results give insights into even clinical problems (ie Menieres disease and betahistine treatment….. and its varying effects).

I find this small study both well done and pedagogically written, including basic inner ear anatomy and function. It is in my opinion a useful contribution.

On line 226 I think a word is missing.

see above

Author Response

We thank the Reviewer for taking the time to read and review our manuscript. Reviewer’s constructive criticism is greatly appreciated, and implementing the pitched ideas improved the readability and quality of our work.

Below you will find the reply to the comment made.

On line 226 I think a word is missing.

This sentence was revised and reads now:

Earlier work in the semicircular canals of the frog showed that betahistine significantly reduced the resting, but not the afferent, discharge.

Reviewer 4 Report

In the article "Histamine and its receptors in the mammalian inner ear: a scoping review", Kong, Domareck and Szczepek give an overview of studies on the histaminergic signaling in the inner ear of rodents. The review will be of interest to other researchers interested in signaling systems in the inner ear and their relevance to understand development, physiology and pathology.

General comment:
Since most articles reviewed are from work on rodents, the authors may consider substituting mammalian with murine or rodent in the title and throughout the text. This can also be discussed as a line of future research, since more studies in humans and other key mammalian species (e.g., non human primates and others) may be needed to generalize over the whole group of mammals.

Introduction:
The introduction could be improved. In particular the description of the inner ear as a "complex sensory organ" can be clarified. The inner ear contains multiple sensory organs, some specialized for vestibular function and others for acoustic function. In mammals there is agreement that there is one acoustic organ, and several vestibular organs. Authors can clarify their initial description (lines 54-55) of the inner ear consisting of the cochlea, the vestibule and the endolymphatic sac, with their secondary definition where the vestibular system is described to comprise the semicircular canals and the vestibule (line 65). Please clarify how many acoustic and how many vestibular organs form part of the inner ear in mammals. This is relevant to understand the results section and figure 1.

Please clarify what an "orphan topic" refers to (line 78). Authors may also consider that the topics of immune function, inflammation and vascular control in the inner ear have been reviewed and researched in recent years.

Please clarify what is the relevance to the introduction for the statement in lines 80-81. I gather from the text in lines 81-83 that mast cells may be considered a source of histamine in the inner ear, but the biological context is missing from the introduction. A more clear explanation of the significance of the research finding on mast cells and its impact for understanding histaminergic signaling is needed for the non-specialist. Note that most of this information was included in the abstract, but is missing from the introduction.

Materials and Methods:
Systematic and clear approach. What was the rationale to select 2000-2023 as the review period?

Results:
The results section has one figure and two tables. The description of findings gives a good overview, but the table is long and does not intuitively match the organization of the discussion. One potential solution might be to include a Venn diagram summarizing the main areas covered by the studies (anatomical/cellular localization, vascular permeability and electrophysiology).

Discussion:
It is not clear from the present version how well the murine data matches the human data. The authors also identify development as a potential new line of research. Why was development not included in the original criteria for searching the databases?

Author Response

We thank the Reviewer for taking the time to read and review our manuscript. Reviewer’s constructive criticism is greatly appreciated, and implementing the pitched ideas improved the readability and quality of our work.

Below you will find point-to-point replies to all comments made.

General comment:
Since most articles reviewed are from work on rodents, the authors may consider substituting mammalian with murine or rodent in the title and throughout the text. This can also be discussed as a line of future research, since more studies in humans and other key mammalian species (e.g., non human primates and others) may be needed to generalize over the whole group of mammals.

The comment is appreciated, and we have considered revising the title. However, given that we included two publications that studied humans (11% of the included studies), the current title more accurately reflects the content of our review.

Introduction:
The introduction could be improved. In particular the description of the inner ear as a "complex sensory organ" can be clarified. The inner ear contains multiple sensory organs, some specialized for vestibular function and others for acoustic function. In mammals there is agreement that there is one acoustic organ, and several vestibular organs. Authors can clarify their initial description (lines 54-55) of the inner ear consisting of the cochlea, the vestibule and the endolymphatic sac, with their secondary definition where the vestibular system is described to comprise the semicircular canals and the vestibule (line 65). Please clarify how many acoustic and how many vestibular organs form part of the inner ear in mammals. This is relevant to understand the results section and figure 1.

Following this suggestion, the entire paragraph was rewritten and now reads as follows:

The vestibular system is responsible for sensing and processing information about the position and movement of the head and body in space and maintaining balance and coordination during the movement [32]. The peripheral vestibular organs are located bilaterally in the inner ear. They consist of two otolithic organs (utricle and saccule) and three semicircular canals (anterior, posterior, and horizontal), the former sensing linear acceleration, such as head movement or gravity, and the latter sensing rotational acceleration [33]. The sensory nerve epithelium in the utricle and saccule is the macula, and in the semicircular canals is the crista ampullaris [32]. Both structures contain vestibular hair cells, which release glutamate upon depolarization, stimulating the vestibular ganglion's afferent nerves. The vestibular ganglion and the cochlear spiral ganglion neurons form the eighth cranial nerve.

Please clarify what an "orphan topic" refers to (line 78). Authors may also consider that the topics of immune function, inflammation and vascular control in the inner ear have been reviewed and researched in recent years.

The sentence in line 78 was revised, “orphan topic” removed, and the information suggested was added. The passage reads now as follows:

In recent years, immune function, inflammatory processes, and vascular control of the inner ear have been investigated and reviewed [34-37]. However, the specific topic of histamine and its signaling in the cochlea or vestibular organs remains scarcely addressed in the literature.

Please clarify what is the relevance to the introduction for the statement in lines 80-81. I gather from the text in lines 81-83 that mast cells may be considered a source of histamine in the inner ear, but the biological context is missing from the introduction. A more clear explanation of the significance of the research finding on mast cells and its impact for understanding histaminergic signaling is needed for the non-specialist. Note that most of this information was included in the abstract, but is missing from the introduction.

To address this concern, we have expanded this paragraph to include the following:

Mast cells are the major source of histamine in the body, along with basophils, gastric parietal cells, and the central nervous system [43]. Upon activation, mast cells degranulate and release a number of immuno- and neuromodulatory compounds, including histamine [44,45]. Some conditions necessary for mast cell activation have already been described in the inner ear, including IgE antibody transcytosis across the blood-labyrinth barrier [46] and the presence of substance P [47]. However, more research is needed to understand the relationship between the presence of mast cells in the inner ear, their mode of activation, potential histamine release and its consequences in health and disease, such as mastocytosis or IgE-mediated diseases. Clinical evidence suggests an association between elevated numbers of mast cells and inner ear disorders [48,49]. Moreover, experiments demonstrated that Meniere's disease-like symptoms (attacks of nystagmus and hearing loss) can be induced by experimental induction of type I allergy in the endolymphatic sac of guinea pigs [50].

Materials and Methods:
Systematic and clear approach. What was the rationale to select 2000-2023 as the review period?

We appreciate that comment. Originally, we performed a search that was not time-restricted. However, the number of articles published before 2000 that met the inclusion criteria was small (n=3), one publication was not available in the full version, and the findings of two other papers were included in subsequent publications of the same groups, making it redundant.

Results:
The results section has one figure and two tables. The description of findings gives a good overview, but the table is long and does not intuitively match the organization of the discussion. One potential solution might be to include a Venn diagram summarizing the main areas covered by the studies (anatomical/cellular localization, vascular permeability and electrophysiology).

We added the requested diagram with the following comment:

The main areas covered by the included studies were anatomical/cellular localization of histamine receptors, vascular permeability, and electrophysiology. However, only one of the studies covered more than one area, combining the anatomical and electrophysiological approaches (Figure 2). One publication, which was not included in the Venn diagram [52], dealt with cochlear metabolomics.

Discussion:
It is not clear from the present version how well the murine data matches the human data. The authors also identify development as a potential new line of research. Why was development not included in the original criteria for searching the databases?

We were indeed interested in studies that focused on histamine in inner ear development. Since the search terms were very generous [histamine AND ((inner ear) OR (cochlea*))], we thought we would identify such studies, but this was not the case. We performed an additional search, but it returned no hits.

To address that, we added the following statement to the Discussion:

However, the expression pattern of histamine receptors in the developing cochlea and the physiological role of these receptors remain unknown, and we did not identify any studies on this topic.